# Classification of Environmental Strains from Order to Genus Levels Using Lipid and Protein MALDI-ToF Fingerprintings and Chemotaxonomic Network Analysis

**DOI:** 10.3390/microorganisms10040831

**Published:** 2022-04-17

**Authors:** Marceau Levasseur, Téo Hebra, Nicolas Elie, Vincent Guérineau, David Touboul, Véronique Eparvier

**Affiliations:** 1CNRS, Institut de Chimie des Substances Naturelles (ICSN), UPR 2301, Université Paris-Saclay, Avenue de la Terrasse, 91 198 Gif-sur-Yvette, France; marceau.levasseur@cnrs.fr (M.L.); teo.hebra@cnrs.fr (T.H.); nicolas.elie@cnrs.fr (N.E.); vincent.guerineau@cnrs.fr (V.G.); 2Laboratoire de Chimie Moléculaire (LCM), CNRS UMR 9168, École Polytechnique, Institut Polytechnique de Paris, Route de Saclay, CEDEX, 91 128 Palaiseau, France

**Keywords:** environmental microorganisms, classification, mass spectrometry, MALDI, molecular network

## Abstract

During the last two decades, MALDI-ToF mass spectrometry has become an efficient and widely-used tool for identifying clinical isolates. However, its use for classification and identification of environmental microorganisms remains limited by the lack of reference spectra in current databases. In addition, the interpretation of the classical dendrogram-based data representation is more difficult when the quantity of taxa or chemotaxa is larger, which implies problems of reproducibility between users. Here, we propose a workflow including a concurrent standardized protein and lipid extraction protocol as well as an analysis methodology using the reliable spectra comparison algorithm available in MetGem software. We first validated our method by comparing protein fingerprints of highly pathogenic bacteria from the Robert Koch Institute (RKI) open database and then implemented protein fingerprints of environmental isolates from French Guiana. We then applied our workflow for the classification of a set of protein and lipid fingerprints from environmental microorganisms and compared our results to classical genetic identifications using 16S and ITS region sequencing for bacteria and fungi, respectively. We demonstrated that our protocol allowed general classification at the order and genus level for bacteria whereas only the Botryosphaeriales order can be finely classified for fungi.

## 1. Introduction

Microorganisms are one of the first sources of enzymes and metabolites with wide applications in the industrial or health fields [1,2]. Their identification is necessary to access all the knowledge that has been shared by the scientific community thanks to the literature and open-source databases [3,4,5]. Since Woese’s work in 1977, the identification of microorganisms has been based on molecular biology methods, in particular the sequencing of the gene coding for 16S rRNA, in the case of bacteria, or 18S rRNA or internal transcribed spacers (ITS) for fungi, and more recently, whole genome sequencing [6,7,8]. However, these methods still require careful sample preparation and the latter one has a high cost that does not favor its systematic application when dealing with large collections [6]. Furthermore, the taxonomic resolution offered by 16S rRNA gene sequencing is not sufficient to identify certain bacterial strains, unless third generation sequencers and denoising algorithms are used in combination [9]. As for fungi, ITS is not a universal barcode because of sequence redundancy between different species-rich genera [10].

Thus, the identification of microorganisms by MALDI-ToF mass spectrometry has become widely democratized during the last two decades; in the field of clinical microbiological diagnosis and agri-food industry, and finally for environmental studies [11,12,13,14]. MALDI-ToF mass spectrometry has proven to be a robust, rapid, and low-cost method for isolate identification or dereplication [15,16,17,18]. This is why one of the current methodologies used for human pathogenic strains consists in implementing private databases with new reference spectra. This method is experimentally and analytically tedious, in particular because the identification of redundant profiles is based on global similarity analyses by hierarchical clustering. Moreover, visually analyzing these dendrograms involves tedious human intervention due to the difficulty of reading and interpreting them as the amount of data analyzed increases [19].

Furthermore, MALDI-ToF mass spectrometry identification of representatives of a cultivable environmental microbial community proves to be more difficult than identification of a pathogenic microorganism due to the high proportion of only pathogenic microorganisms in the available databases [12,13]. Thus, fingerprint discrimination by detecting specificities could allow a better dereplication [20]. Among the chemo-informatic pipelines developed to allow classification and/or dereplication of metabolomics data, molecular networking is becoming more and more popular [21,22,23,24]. Its principle is based on the postulate that structurally close molecules share the same fragmentation pattern in mass spectrometry (MS). Although the construction of molecular networks has been used only for MS² data, it is also possible at MS^1^ level, according to a direct comparison of MS^1^ spectra [21,25,26]. Recently, Dumolin et al. demonstrated the robustness of this methodology through the creation of SPeDe; an open-source software for high-throughput dereplication of isolate fingerprints from MALDI-ToF MS analyses [27,28].

In this study, we proposed a methodology based on global similarity recognition of protein and lipid fingerprints from environmental bacteria or fungi where the spectrum/taxonomy match of the isolate is confirmed by sequencing of DNA loci (16S for bacteria and ITS for fungi) correlative to the identity. We validated our in silico analysis methodology by using open-source data, made available by the Robert Koch Institute (RKI, 13 orders, 38 genera, 568 fingerprints), through an R script and MetGem software [25]. Finally, we created a dataset to dereplicate bacterial (9 orders, 49 genera, 138 isolates) and fungal (21 orders, 51 genera, 230 isolates) environmental isolates, from the Bank of Natural Substances & Biodiversity (BNSB, CNRS-ICSN, Gif-sur-Yvette, France, https://icsn.cnrs.fr/en/platforms/strain-library (accessed on 1 January 2021) by analyzing their protein and lipid fingerprint by MALDI-ToF MS.

## 2. Results

### 2.1. Proof of Concept: t-SNE Algorithm Clusters Bacterial Protein Fingerprints in a Taxonomy-Consistent Manner

In order to evaluate the functionality of our methodology, we first used a part of the RKI dataset with the aim of comparing the global similarity of bacterial protein fingerprints. The objective was to know if the t-SNE algorithm used by the MetGem software was capable of clustering MALDI-ToF mass spectra, corresponding to protein fingerprints of microorganisms (Figure 1), in order to classify them according to their taxonomy. The t-SNE algorithm is a statistical method for visualizing high-dimensional data that captures local similarities while attempting to preserve the integrity of global structures. For that, we hijacked some MetGem functionalities in order to use the t-SNE algorithm on protein fingerprints from the RKI database. In fact, MetGem is a software able to build molecular networks and is a powerful dereplication tool in the field of metabolomics [25]. Here, MS^1^ data are visualized following a pre-processing of data under a home-made script in R (for more details, see Materials and Methods, Section 4.6 and Appendix A). In this dataset, we have performed t-SNE visualization of 568 fingerprints belonging to 166 different bacterial strains. Each protein fingerprint of an identified clinical isolate was represented by a node (Figure 2).

This method led to the data aggregation into twelve clusters (Figure 2). First clusters, named A and B, are composed of Enterobacterales in which are present the genera *Proteus*, *Serratia*, *Yersinia* (61 fingerprints—49.2% of Enterobacterales), and *Citrobacter*, *Enterobacter*, *Escherichia*, *Klebsiella*, *Proteus*, *Salmonella*, *Serratia*, and *Shigella* (46 fingerprints—37.1% of Enterobacterales), respectively. Genera not present in clusters A and B are *Edwardsiella*, *Enterococcus*, and *Vibrio* which are dispersed to the north of the network or are isolated nodes (15 fingerprints—12.1% of Enterobacterales). Cluster C (Figure 2) is only composed of the genus *Brucella* belonging to the order Rhizobiales (28 fingerprints—87.5% of Rhizobiales). The last genus of this order, i.e., *Ochrobactrum*, is located south of cluster C (4 fingerprints—12.5%). Then, cluster D (in pink, Figure 2) includes the majority of Thiotrichales, of which only the genus *Francisella* is represented (20 fingerprints—83.3% of Thiotrichales). Fingerprints corresponding to the species *Francisella guangzhouensis* are at the opposite of cluster D (4 fingerprints—16.7%). Afterwards, clusters E, F, G, H, I, J (Figure 2) are composed of Burkholderiales. Among them, the genus *Burkholderia* is present only in clusters F, G, H, I, and in their periphery (134 fingerprints—80.7% of Burkholderiales). Clusters E and J includes species of the genera *Pandoraea* and *Ralstonia*, respectively (24 fingerprints—14.5%). The last nodes with fingerprints from Burkholderiales are scattered in the network; east of its center (genus *Achromobacter*) or close to the J cluster (genera *Oligella* and *Xenophilus*) (8 fingerprints—4.8% of Burkholderiales). Finally, clusters K and L (Figure 2) are solely composed of fingerprints from isolates belonging to the genus *Bacillus* (91 fingerprints—53.8% of Bacillales). The last fingerprints from Bacillales are distributed in the center of the network or in the south region and belong to the genera *Bacillus*, *Lysinibacillus*, *Paenibacillus*, and *Staphylococcus* (78 fingerprints—46.2% of Bacillales).

The protein fingerprints belonging to the rare orders (55 fingerprints), i.e., Actinomycetales, Aeromonadales, Campylobacterales, Lactobacillales, Pseudomonadales, Rhodobacterales, Rhodospirillales, Xanthomonadales, of this dataset are scattered in the network (Figure 2) but all replicates of an isolate are close in space confirming a high robustness of the sample preparation and MALDI analysis.

In order to verify the robustness of chemotaxonomic identifications by fingerprint comparison, we added the protein fingerprints of our environmental bacterial strains to this first network.

### 2.2. Robustness of Chemotaxonomic Resolution by Adding Fingerprints of Environmental Isolates to the RKI Dataset

This new analysis was also obtained by visualizing the protein fingerprints of the RKI pathogenic bacteria and those of environmental bacteria through t-SNE algorithm (Figure 2). For this purpose, the initial dataset was supplemented with 138 fingerprints of bacteria whose taxonomic affiliation was previously determined by comparing the DNA sequences, encoding 16S rRNA, with those of NCBI. No new order was added to the dataset, majority orders are the same as for the RKI, i.e., Bacillales, Burkholderiales, and Enterobacterales.

Of these 138 fingerprints, 81 nodes (58.7%) of our dataset join the RKI dataset or form a new Enterobacterales cluster (Cluster M in Figure 3). Thus, 44 nodes are isolated (31.9%) and 13 (9.4%) are found in clusters consisting of protein fingerprints belonging to bacteria of another taxonomic order. In addition, 3 fingerprints of the genus *Vibrio* initially isolated in Figure 2 are clustered with Burkholderiales and Enterobacteriales fingerprints from the BSNB, east of the network.

In order to refine our identification by MALDI-ToF fingerprinting, we have chosen to look in parallel at the lipid fingerprints of our different strains.

### 2.3. Differentiation of Environmental Bacteria by MALDI-ToF MS Lipid Fingerprint Analysis

The purpose of this analysis was to evaluate whether lipid fingerprinting analysis of environmental microorganisms provided better differentiation than the conventional protein fingerprinting approach (Figure 4). Thus, 138 lipid fingerprints were performed on the BNSB microorganisms including the same bacteria as those for which the protein fingerprint dataset had been performed (Figure 3).

Our chemo-informatics pipeline led to the clusterization of the major part of the dataset. Cluster N is composed mainly of Bacillales (24 fingerprints—65% of Bacillales), then Burkholderiales (1 fingerprint—2.8% of Burkholderiales) and Rhizobiales (1 fingerprint—11.1% of Rhizobiales). Cluster O is mainly composed of Rhizobiales (8 fingerprints—88.9%), then Pseudomonadales (1 fingerprint—50% of Pseudomonadales). Cluster P is composed of two main orders, i.e., Enterobacterales (18 fingerprints—56.3% of Enterobacterales) and Burkholderiales (17 fingerprints—47.2% of Burkholderiales), then Lactobacillales (2 fingerprints—100% of Lactobacillales). Cluster Q1 contains PEGs inducing clustering of lipid fingerprints from different isolates. The set of isolated nodes contains 7 Actinomycetales, 11 Bacillales, 15 Burkholderiales, 11 Enterobacterales, 1 Pseudomonadales, 1 Rhodospirillales, and 1 Xanthomonadales.

As this methodology was initially applied to our environmental bacteria of our strains collection, we wanted to evaluate if data from fungi could also be efficiently classified.

### 2.4. Differentiation of Environmental Fungi by MALDI-ToF MS Protein Fingerprint Analysis

Previously, we demonstrated the robustness of our methodology by analyzing protein or lipid fingerprints of environmental bacteria. Here, we applied the same methodology to protein fingerprints of environmental fungi. To this end, we first compared the protein fingerprints of 230 fungi from our collection. The resulting chemotaxonomic network is presented in Figure 5. In this network, several clusters have been identified.

The cluster R is composed of 18 Glomerellales (32.7% of Glomerellales) of which 16 are *Colletotrichum gloeosporioides* (52.9%) against 1 *C. theobromicola* and 1 *C. siamense*. Finally, this cluster also contains 2 Xylariales (5% of Xylariales) protein fingerprints. Cluster S is mainly composed of Diaporthales (15 fingerprints—71.4% of Diaporthales), then of Xylariales (1 fingerprint—2.5% of Xylariales) and Capnodiales (1 fingerprint—20% of Capnodiales). Cluster T is composed of 10 Xylariales (25% of Xylariales), 4 Glomerellales (0.7% of Glomerellales), and 1 Cantharellales (50% of Cantharellales). Group U is mainly composed of Glomerellales (10 fingerprints—18.2% of Glomerellales) of which the majority species is *Colletotrichum theobromicola* (66.7%). It also contains 1 Eurotiales (4.3%) and 1 Xylariales (2.5%). Finally, cluster V is mainly composed of Botryosphaeriales (12 fingerprints—75%), then of Hypocreales (1 fingerprint—3.3%) and Xylariales (1 fingerprint—2.5%). Finally, the set of isolated nodes contains 1 Capnodiales, 1 Chaetothyriales, 3 Diaporthales, 18 Eurotiales, 9 Glomerellales, 11 Hypocreales, 1 Microascales, 1 Microthyriales, 1 Mucorales, 1 Pleosporales, 2 Russulales, 2 Saccharomycetales, 4 Sordariales, 1 Venturiales, and 12 Xylariales.

Lipid fingerprinting of the same fungal strains was also performed.

### 2.5. Differentiation of Environmental Fungi by MALDI-ToF MS Lipid Fingerprint Analysis

The protein fingerprint analyses performed resulted in the network shown in Figure 6. Several clusters that appeared to be specific to some orders have been identified.

First, cluster X is mainly composed *of* Eurotiales (5 fingerprints—21.7% of Eurotiales) and Hypocreales (10 fingerprints—33.3% of Hypocreales), then of Sordariales (1 fingerprint—11.1% of Sordariales) and Xylariales (2 fingerprints—5% of Xylariales). Then, cluster Y is composed of Cantharellales (2 fingerprints—100% of Cantharellales), Capnodiales (3 fingerprints—60% of Capnodiales), Hypocreales (1 fingerprint—3.33% of Hypocreales), Russulales (3 fingerprints—50% of Russulales), Venturiales (1 fingerprint—50% of Venturiales), and Xylariales (12 fingerprints—30% of Xylariales). Cluster Z is only composed of Botryosphaeriales (12 fingerprints—75% of Botryosphaeriales) and all fingerprints of isolates belonging to this cluster are *Endomelanconiopsis endophytica*. The fingerprints of the other representatives of the order Botryopshaeriales are *Guignardia mangiferae* scattered in the central group or are isolated nodes. Finally, cluster Q2 contains PEGs inducing clustering of lipid fingerprints from different isolates.

The set of isolated nodes contains 2 Botryosphaeriales, 1 Chaetothyriales, 1 Eurotiales, 4 Glomerellales, 1 Hypocreales, 1 Microthyriales, 5 Mucorales, and 4 Xylariales.

With the exception of the Botryosphaeriales, the different orders do not cluster specifically in the constructed network.

## 3. Discussion

The identification and dereplication of environmental isolates is a complex objective due to the analysis of biological objects, i.e., DNA, protein, lipid, reflecting the taxonomic affiliation of the organisms studied. Today, this methodology is mostly conducted by the dereplication of protein fingerprints obtained by MALDI-ToF mass spectrometry. While this methodology is widely considered to be robust in the clinical context, it is difficult to access the identity of isolates from environmental samples due to the lack of fingerprints of organisms not pathogenic to humans in current databases.

In this study, a standardized workflow for extracting lipids and proteins from environmental microorganisms at medium throughput in a single tube of reaction medium was proposed. Then, the pre-processing of the fingerprints from the MALDI-ToF MS analyses using a “home-made” R script and the analysis of the resulting data using the MetGem software was done. Finally, the reliability of the taxonomic affiliation of the organisms was ensured by the sequencing of the loci corresponding to the 16S rDNA for bacteria or the ITS for fungi.

In a first step, we validated our data analysis method by retrieving a portion of the protein fingerprints of highly pathogenic microorganisms from the free RKI dataset (Figure 2). The t-SNE algorithm is able to group the RKI protein fingerprints according to the taxonomic affiliation of the studied organisms based on the order for Enterobacterales according to 2 clusters (A and B in Figure 2), Burkhorderiales according to 6 clusters (E, F, G, H, I & J in Figure 2) and Bacillales according to 2 clusters (K and L in Figure 2). If the taxonomic resolution of this representation is validated by the spatial proximity of the nodes corresponding to the same order or genus, the results obtained are biased by the diversity and representativeness of the protein fingerprints present in the analyses. This problem is often found in environmental sampling campaigns, due to the environmental matrices used, the culture conditions of the organisms and therefore, their physiology in laboratory, without forgetting the sample preparation methodology [12,13,19,29,30].

As a second intention, we wanted to know if some protein fingerprints could be implemented and correlated with those already existing in the RKI dataset. Therefore, we acquired protein fingerprints of environmental bacteria maintained at ICSN to adjoin those of the RKI (Figure 3). Despite affiliation errors (9.4%) and lack of matches (31.9%), 58.7% of the acquired protein fingerprints were grouped in their corresponding taxonomic order or genus. Once again, the most represented orders were: Bacillales (29.8%), Burkhorderiales (29.2%), and Enterobacterales (21.8%). This demonstrates the possibility of building bacterial protein fingerprint annotation networks by implementing data from the scientific community in a free manner using our workflow. In contrast, the taxonomic resolution of fungal protein fingerprints in our analysis is less efficient than that performed on our bacterial dataset. Despite a greater diversity and a larger dataset (21 orders including 51 fungal genera against 9 orders including 24 bacterial genera), few orders are annotatable except for the Glomerellales which are distributed, mostly, in 2 clusters, R corresponding to the species *C. gloeosporioides* and U to *C. theobromicola*, and the Botryosphaeriales constituting the specific cluster V. Moreover, Diaporthales can be annotated according to our approach because of their majority distribution (71.4% in cluster S, Figure 4).

Today, biotyping is widely used for the dereplication of protein fingerprints, but until now, few studies have focused on the same methodology for lipid fingerprints, which are mainly used to identify fungi [30,31,32]. Here, we analyzed the lipid fingerprints of the same bacteria as those used in the previous analysis (Figure 3). Again, it is possible to annotate the identity of the Bacillales (cluster N in Figure 4). However, additional data should be acquired to verify this statement on the representatives belonging to the orders Burkholderiales and Enterobacterales because their lipid fingerprints are grouped in the same cluster (cluster P, Figure 4). On the other hand, the analysis of lipid fingerprints of fungi shows a low taxonomic resolution, due to the formation of only one specific cluster of Botryosphaeriales (cluster Z in Figure 6). However, Stübiger et al. had also observed this phenomenon on a dataset of lower diversity (genera *Aspergillus*, *Penicillium*, *Saccharomyces*, and *Trichoderma*) where commercial databases (Bruker (Bremen, Germany) and Biomérieux (Marcy-l’Étoile, France)) did not satisfy the concordances between lipid fingerprints and taxonomic affiliations [32].

## 4. Materials and Methods

### 4.1. Microorganisms and Cultures

The majority of studied microorganisms are from French Guiana and were studied in previous research projects to explore their metabolome. Briefly, our environmental strain collection includes endophytes, insect-associated microorganisms and entomo- or phytopathogens [33,34,35,36]. All microorganisms were isolated on Potato Dextrose Agar (PDA, Condalab, Madrid, Spain) medium at 28 °C and stored at −80 °C in a solution of water and glycerol (20:80). Finally, their phenotypes were recorded in the Bank of Natural Substances & Biodiversity (BNSB, ICSN-CNRS, Gif-sur-Yvette, France, https://icsn.cnrs.fr/en/platforms/strain-library (accessed on 1 January 2021). The microorganisms were collected with the following ABS authorizations: ABSCH-IRCC-EN-248781-1; ABSCH-IRCC-EN-248782-1 and ABSCH-IRCC-EN-245916-1 or before 2010.

### 4.2. Identification of Isolates

The identification process of the bacteria is performed by amplification of a portion of the gene coding for 16S rRNA (27-F: 5′-AGAGTTTGATCCTGGCTCAG-3′ and 1492-R: 5′-GGTTACCTTGTTACGACTT-3′) by PCR and then sequencing the amplicons (27-F: 5′-AGAGTTTGATCCTGGCTCAG-3′ and 907-R: 5′-CCGTCAATTCCTTTGAGTTT-3′). The same applies to fungal identifications (primers used for PCR and sequencing: ITS1-F: 5′-AGGAGAAGTCGTAACAAGGT-3′ and ITS4-R: 5′-TCCTCCGCTTATTGATATGC-3′). We use the BLASTn algorithm to compare the obtained sequences to those present on the NCBI site. If the local alignments of the sequences obtained by sequencing were significant, then the identity of the microorganism corresponded to its closest relative.

### 4.3. Protein and Lipid Extractions

Unless otherwise stated, all solvents used are of analytical grade. Incubation times differ according to taxonomic affiliation. Thus, proteins and lipids are extracted as soon as colonies appear, i.e., 1 to 3 days for bacteria and 7 days for fungi. We tested 3 protein extraction protocols [31,37,38] and 2 lipid extraction protocols [32,39] before using the following (data not shown): extraction protocol is adapted from Cassagne et al. and Stübiger et al. [31,34], and consists of introducing 3 to 5 bacterial colonies or a mycelium square (≈5 mm²) in a sterile 2 mL tube containing 300 µL H_2_O Milli-Q^®^, then, 200 µL of methanol (MeOH, Sigma-Aldrich, Saint Quentin Fallavier, France) and 1000 µL of methyl *tert*-butyl ether (MTBE, Sigma-Aldrich, Saint Quentin Fallavier, France). The resulting biphasic solution is mixed thoroughly during 1 min and then allowed to settle at room temperature (≈20 °C) for 5 min or until the two distinct phases reform. Decantation is prolonged overnight at −4 °C. Then, cell lysates are centrifuged at 13,000 rpm during 15 min and the upper organic phase is introduced into a 10 mL glass tube in order to be dried in a rotary evaporator under reduced pressure to obtain a dried lipid sample.

After the recovery of the organic phase, 900 µL of 100% ethanol (EtOH, Sigma-Aldrich, France) is introduced in the lower aqueous phase and homogenized. After 5 min centrifugation at 13,000 rpm, supernatant is discarded and residual pellet is dried at room temperature during 30 min. Then, the pellet is incubated 5 min in 80 to 160 µL of 70% formic acid (FA, Sigma-Aldrich, Saint Quentin Fallavier, France). Finally, an equivalent volume of 100% acetonitrile (ACN, Sigma-Aldrich, Saint Quentin Fallavier, France) is added before centrifugation (13,000 rpm, 5 min) and the supernatant, containing proteins, is conserved in a 10 mL glass tube, awaiting analysis.

Each series of extraction is carried out with an extraction control containing only PDA medium and another containing no biological material. Samples are conserved at −20 °C awaiting analysis.

### 4.4. MALDI-ToF Sample Preparation

Each protein sample is analyzed in duplicate for which 1 µL is deposited on a spot of polished steel plate (MTP384 polish steel target, Bruker Daltonics GmbH, Bremen, Germany), then mixed with 1 µL of matrix solution and air-dried. The matrix solution for protein analysis is prepared before each series of analyses and is composed of 20 mg of α-cyano-4-hydroxycinnamic acid (CHCA, Sigma-Aldrich, France), partially solubilized in a solution of 1 mL of ACN/H_2_O/trifluoroacetic acid (50/50/2.5, *v*/*v*/*v*).

Each dried lipid sample is solubilized in 40 to 80 µL of CHCl_3_/MeOH (2/1, *v*/*v*), then 1 µL of the solubilized samples is diluted in a matrix solution for lipid analysis in a 1:5 ratio. The matrix solution for lipid analysis is prepared before each series of analyses and is composed of 20 mg of 2,5-dihydroxybenzoic acid (DHB, Sigma-Aldrich, France) solubilized in 500 µL of tetrahydrofuran (THF, Sigma-Aldrich, France). Finally, 1 µL of this solution is spotted on polished steel plate and left to air-dry.

### 4.5. MALDI-ToF Mass Spectrometry

Protein and lipid mass spectra are acquired on a MALDI-ToF/ToF UltrafleXtrem (Bruker Daltonics GmbH, Bremen, Germany) equipped with a 337 nm pulsed nitrogen laser in the linear and reflectron mode using delayed ion extraction, in positive ion mode and by accumulating, at least twice on each replicate, 1000 single laser shots. Acquisition parameters for protein analysis are the following: linear mode, delay: 250 ns, ion source voltage 1:20 kV, ion source voltage 2:18.5 kV, and mass range: *m*/*z* 2000 to 20,000. Acquisition parameters for lipid analysis are the following: reflectron mode, delay: 140 ns, ion source voltage 1:20 kV, ion source voltage 2:18 kV, reflectron analyzer 1:21.5 kV, reflectron analyzer 2:11 kV, and mass range: *m*/*z* 400 to 2000. The mass spectrometer is externally calibrated using a mixture of proteins (insulin, cytochrome C, myoglobin, and ubiquitin I) in linear mode, and a mixture of polyethylene glycol (PEG, Sigma-Aldrich, Saint Quentin Fallavier, France) in reflectron mode.

### 4.6. Spectra Processing

Mass spectra from the Robert Koch Institute [40] have been sorted so that taxonomic orders or genera are redundant with the taxonomic affiliation of some bacteria studied. Rare orders were deliberately added to the dataset to control the distribution of their corresponding nodes within the constructed molecular networks. In total, 568 spectra were selected, corresponding to 13 taxonomic orders among which: 12 (2.1%) are Actinomycetales, 4 (0.7%)—Aeromonadales, 169 (29.8%)—Bacillales, 166 (29.2%)—Burkholderiales, 4 (0.7%)—Campylobacterales, 124 (21.8%)—Enterobacteriales, 4 (0.7%)—Lactobacillales, 12 (2.1%)—Pseudomonadales, 34 (6%)—Rhizobiales, 4 (0.7%)—Rhodobacterales, 4 (0.7%)—Rhodospirillales, 24 (4.2%)—Thiotrichales, and 7 (1.2%)—Xanthomonadales, i.e., 38 genera (Appendix A).

The bacteria of the BNSB belong to the 9 following orders: 11 (8%)—Actinomycetales, 40 (29%)—Bacillales, 36 (26.1%)—Burkholderiales, 32 (23.2)—Enterobacteriales, 2 (1.4%)—Lactobacillales, 3 (2.2%)—Pseudomonadales, 9 (6.5%)—Rhizobiales, 2 (1.4%)—Rhodospirillales, 3 (2.2%)—Xanthomonadales, i.e., 24 genera (see Appendix A). The combined RKI and BSNB protein fingerprint dataset contains 13 orders, 49 genera, and 706 fingerprints (Figure 3).

Fungi from our collection belong to the 21 following orders: 16 (7%)—Botryosphaeriales, 2 (0.9%)—Cantharellales, 5 (2.2%)—Capnodiales, 1 (0.4%)—Chaetothyriales, 1 (0.4%)—Cystobasidiales, 21 (9.1%)—Diaporthales, 23 (10%)—Eurotiales, 55 (23.9%)—Glomerellales, 30 (13%)—Hypocreales, 2 (0.9%)—Magnaporthales, 4 (1.7%)—Microascales, 1 (0.4%)—Microthyriales, 7 (3%)—Mucorales, 1 (0.4%)—Pleosporales, 1 (0.4%)—Polyporales, 6 (2.6%)—Russulales, 2 (0.9%)—Saccharomycetales, 9 (3.9%)—Sordariales, 1 (0.4%)—Sphaeropleales, 2 (0.9%)—Venturiales, and 40 (17.4%)—Xylariales, i.e., 51 genera (Appendix A).

All spectra are converted in mzXML file format with MSConvert (v. 3.0.20344), a tool from ProteoWizard software [41], then are processed with a home-made script written in R software with MALDIquant and MALDIquantForeign packages [42]. Our methodology is the following: first, mass range of the protein mass spectra is adjusted to be between 3.5–20 kDa in order to avoid the implementation of background noise in our data, then the peaks with a signal to noise ratio ≥6 are selected, the intensities are transformed by applying a square root function, smoothed by Savitzky–Golay method (half-window size = 15) [43], then, baseline removal is conducted by Statistics-sensitive Non-linear Iterative Peak-clipping algorithm (SNIP, 30 iterations), finally, spectra are normalized using Total Ion Current (TIC) method. The last step is to create a Mascot Generic Format (MGF) file that can be interpreted by MetGem software [25]. The code is available in Appendix A and at https://github.com/MarceauLEVASSEURCNRS/20220119_MALDI_mgf (accessed on 19 January 2022).

## 5. Conclusions

Our study proposed an open-source workflow including a standardized protein and lipid extraction protocol, in a tube of reaction medium, as well as a pre-processing of the data under a home-made R script, and an analysis of the fingerprints obtained by the creation of chemotaxonomic networks under the MetGem software. Our results show that this method can be used to discriminate, at medium throughput, bacterial isolates from protein or lipid fingerprints if there are sufficient representatives of a bacterial order or genus. On the other hand, the application of this same process to tropical filamentous fungi remains to be improved because of the sole clustering of orders or genera overrepresented in our data, i.e., Botryosphaeriales and Glomerellales. Moreover, the *sine qua non* condition for the identification of environmental isolates will be the construction of an appropriate spectral database. To our knowledge, this work is the first to focus on the chemotaxonomy of tropical fungi. Thus, our objective is to provide the scientific community with a free and implementable spectral database of tropical microorganisms’ fingerprints with a particular attention to filamentous fungi in order to improve the dereplication processes of environmental strains used following sampling campaigns. These results are encouraging and this method can be used to discriminate two different strains isolated from the same environment and the same host (in the case of symbiotic microorganisms). This first step allows to avoid redundant studies and thus to accelerate the research in the field of natural substances chemistry. Finally, this dereplication based on the calculation of cosine scores enables analysis of large datasets.

## Figures and Tables

**Figure 1 microorganisms-10-00831-f001:**
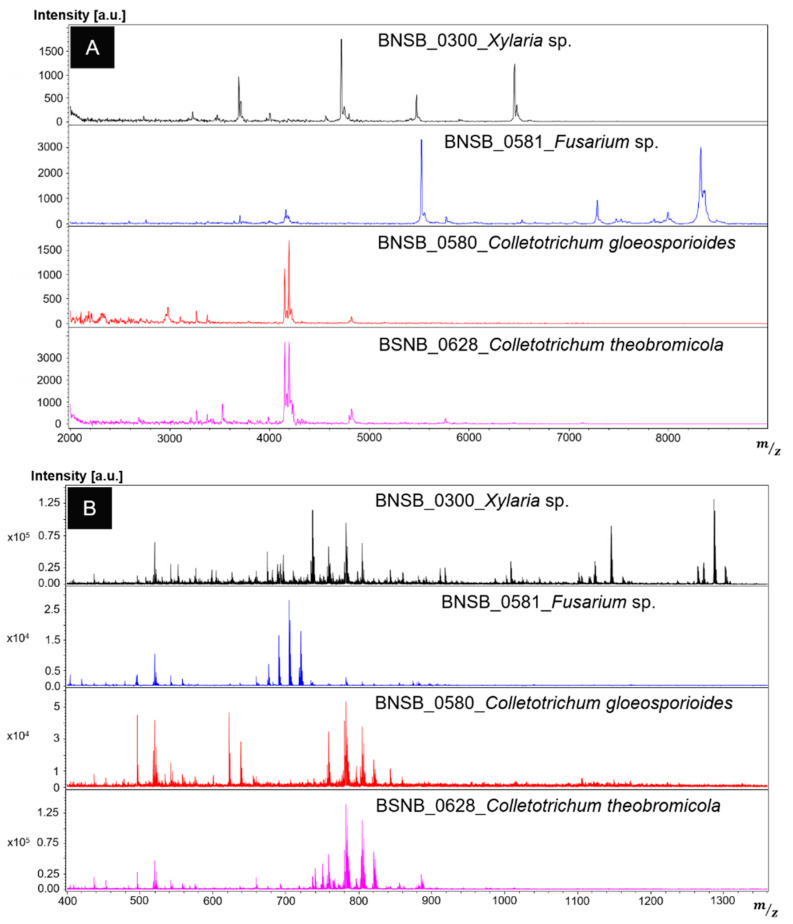
Examples of fungal protein fingerprints (**A**) and fungal lipid fingerprints (**B**) from BNSB.

**Figure 2 microorganisms-10-00831-f002:**
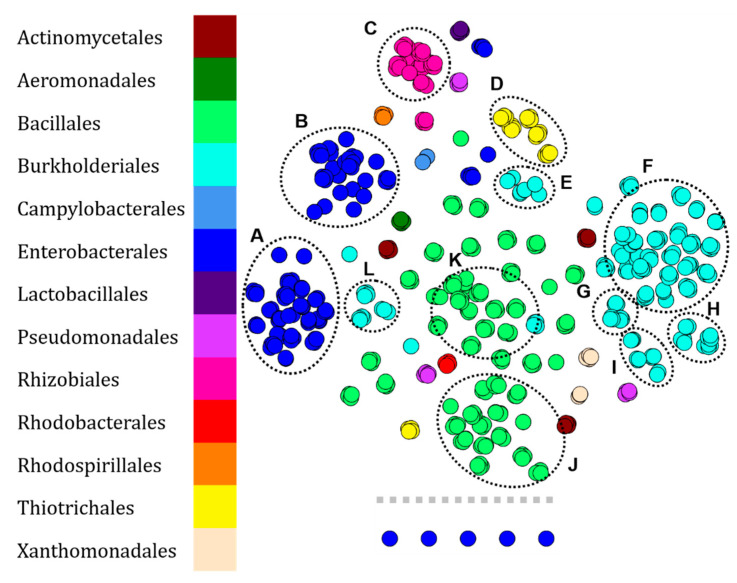
Visualization of 568 protein fingerprints of pathogenic bacteria (166 strains) from a part of the RKI data set by t-SNE. Each node represents the protein fingerprint of a clinical isolate identified at RKI and is colored according to the taxonomic order of the organism. The observed clusters were noted from A to L.

**Figure 3 microorganisms-10-00831-f003:**
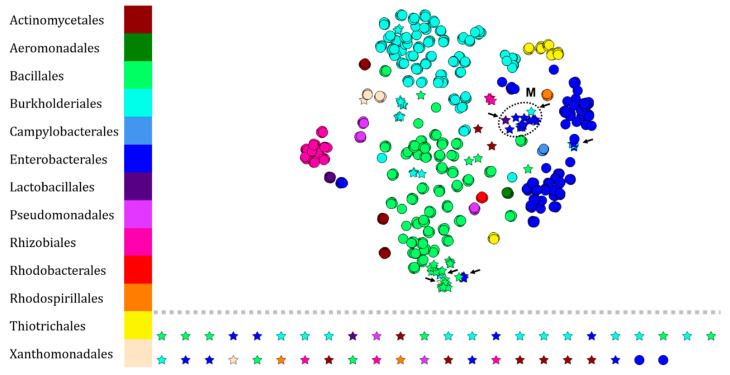
Visualization of 706 protein fingerprints of pathogenic bacteria (166 strains—RKI’s database; •) and environmental bacteria (138 isolates—BNSB’s database; ★) set by t-SNE. Each node represents the protein fingerprint of an isolate and is colored according to the taxonomic order of the organism. Arrows indicate illogical attributions within clusters. The annotated cluster M consists mainly of enterobacteria.

**Figure 4 microorganisms-10-00831-f004:**
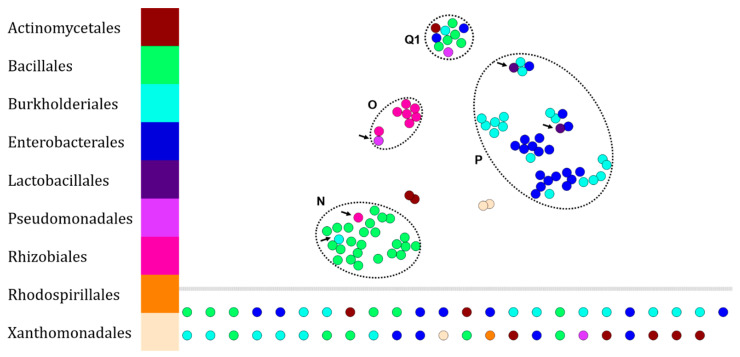
Visualization of 138 lipid fingerprints of environmental bacteria (138 isolates—BNSB’s database) set by t-SNE. Each node represents the lipid fingerprint of an isolate and is colored according to the taxonomic order of the organism. Arrows indicate illogical attributions within clusters (noted Q1, N, O and P).

**Figure 5 microorganisms-10-00831-f005:**
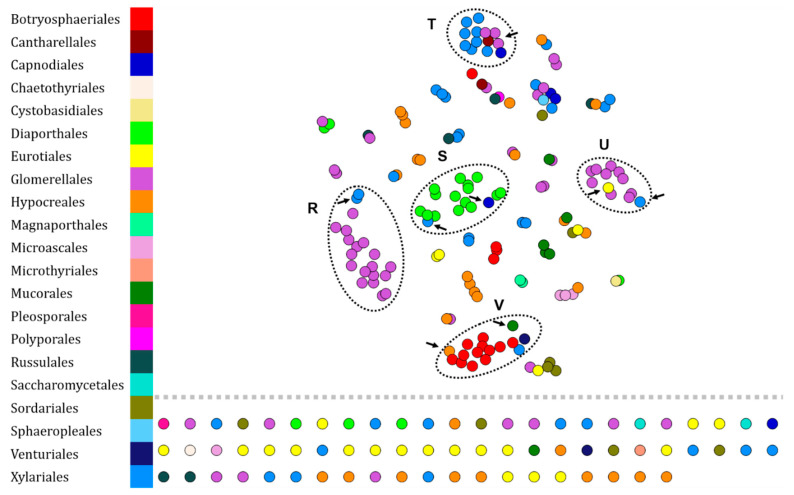
Visualization of 230 protein fingerprints of environmental fungi (230 isolates—BNSB’s database) set by t-SNE. Each node represents the protein fingerprint of an isolate and is colored according to the taxonomic order of the organism. Arrows indicate illogical attributions within clusters (annotated from R to V).

**Figure 6 microorganisms-10-00831-f006:**
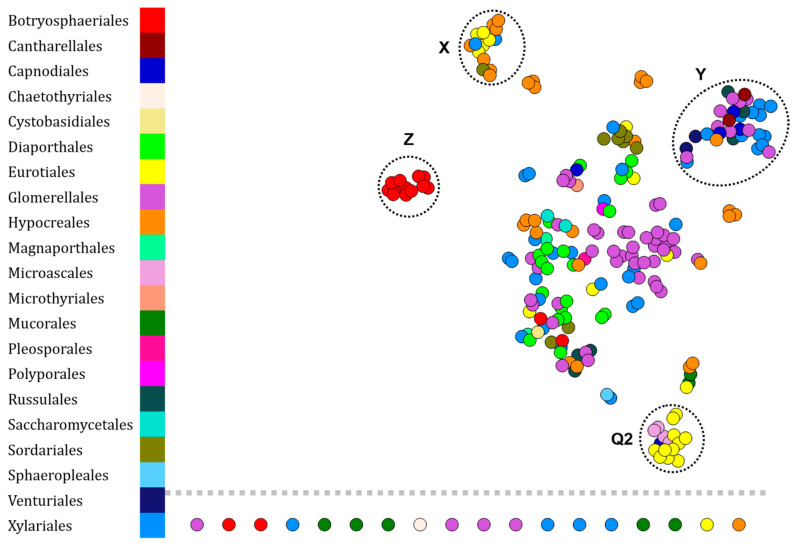
Visualization of 230 lipid fingerprints of environmental fungi (230 isolates—BNSB’s database) set by t-SNE. Each node represents the lipid fingerprint of an isolate and is colored according to the taxonomic order of the organism. Arrows indicate illogical attributions within clusters (annotated Q2, X, Y and Z).

## Data Availability

The data presented in this study are available in the Appendix A.

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
