# Peer review of "Classification of Environmental Strains from Order to Genus Levels Using Lipid and Protein MALDI-ToF Fingerprintings and Chemotaxonomic Network Analysis"

_microorganisms, 2022, doi:10.3390/microorganisms10040831_

Round 1
Reviewer 1 Report
Manuscript proposed by Levasseur and co-workers (microorganisms-1595918) entitled “Classification of environmental strains from order to genus levels using lipid and protein MALDI-ToF fingerprintings and chemotaxonomic network analysis” presents methodology based on global similarity recognition of protein and lipid fingerprints from environmental bacteria or fungi where the spectrum/taxonomy match of the isolate is confirmed by sequencing of DNA loci (16S for bacteria and ITS for fungi) correlative to the identity. In my opinion, presented manuscript is incomplete, needs lots of changes and corrections.
My major comments are presented below.
Major concerns:
- Abstract – clearly present the novelty of the proposed method and its advantages over others used.
- Introduction – present paragraph including description of the other methods used in such kind of analysis
- Result section – what was the total amount of identified proteins and lipids? Were only the most abundant proteins and lipids identified? What similarities between the identified proteins and lipids in used bacteria/fungi were found?
- Result section – lack of statistical analysis. The statistical data should be presented in tables. Statistical data should be discussed in the text.
- Result section – did the Authors analyze the effect of the used matrix on the obtained results? Discussion on the matrix effects in MALDI analysis of lipids and proteins should be presented.
- what was the precision of m/z values determination?
- Materials and method section – par. 4.3 – was the extraction method optimized? How do the Authors know that this method provide good results?
- Materials and method section – par. 4.3, 4.4 – determine the purity of used solvents (methanol, ethanol, chloroform, acetonitrile, formic acid, THF), lack of MS parameters (temperatures, potentials, collision energies, parent ions, method optimization etc.)
- Materials and method section – m/z 2 000 to 20 000 in the case of protein analysis and m/z 400 to 2 000 in the case of lipids analysis was used – why this m/z range was used? Did the Authors optimized the m/z range in the compounds identification? Is there any influence of the m/z range on the identified compounds, their nature and number?
- how sensitive was the analysis?
- how many times was one sample analyzed?
Check and correct the reference style according to the journal guide – positions 40, 41
Make changes in the text.
Check and correct English
Author Response
Reviewer 1
Manuscript proposed by Levasseur and co-workers (microorganisms-1595918) entitled “Classification of environmental strains from order to genus levels using lipid and protein MALDI-ToF fingerprintings and chemotaxonomic network analysis” presents methodology based on global similarity recognition of protein and lipid fingerprints from environmental bacteria or fungi where the spectrum/taxonomy match of the isolate is confirmed by sequencing of DNA loci (16S for bacteria and ITS for fungi) correlative to the identity. In my opinion, presented manuscript is incomplete, needs lots of changes and corrections.
My major comments are presented below.
Major concerns:
- Abstract – clearly present the novelty of the proposed method and its advantages over others used.
- Introduction – present paragraph including description of the other methods used in such kind of analysis
- Result section – what was the total amount of identified proteins and lipids?
In these studies, we performed fingerprint comparisons, i.e. comparison of detected m/z. The number of features is depending on each microorganism. Moreover, as described in the literature, this methodology does not require any lipid or protein identification.
- Were only the most abundant proteins and lipids identified?
The proteins and lipids detected are only the most abundant and/or the most ionized species as described for MALDI fingerprints.
- What similarities between the identified proteins and lipids in used bacteria/fungi were found?
Thank you for your interesting comment. The results are very disparate between bacteria and fungi and the results obtained by combining the analyses of these two resources are not conclusive, as already described in the literature (reference 12 and 13). We therefore did not perform this analysis. Unfortunately, it is impossible to combine the protein and lipid datasets because the parameters used in MetGem are optimized for each type of biomolecule.
- Result section – lack of statistical analysis. The statistical data should be presented in tables. Statistical data should be discussed in the text.
We used chemical similarity scores and not principal component analyses or other statistical tools to order the dataset explaining that there are no statistical tables introduced in the manuscript.
- Result section – did the Authors analyze the effect of the used matrix on the obtained results? Discussion on the matrix effects in MALDI analysis of lipids and proteins should be presented.
Thank you for this comment. For proteins, HCCA is described as the best matrix for MALDI biotyping in the literature (https://journals.asm.org/doi/10.1128/JCM.02399-13 ) whereas DHB is reported as efficient for lipid fingerprinting (https://link.springer.com/article/10.1007/s00249-006-0090-6 ). Our experience in the lab since more than 20 years in MALDI confirmed this approach from literature.
- what was the precision of m/z values determination?
We used a ToF analyzer in the linear mode for the analysis of proteins, leading to a precision around 50 ppm, while around 5 ppm in reflectron mode for the analysis of lipids . The mass spectrometer is externally calibrated with a mixture of proteins (insulin, cyto-chrome C, myoglobin and ubiquitin I), in linear mode, and a mixture of polyethylene glycol (PEG, Sigma-Aldrich, France), in reflectron mode.
- Materials and method section – par. 4.3 – was the extraction method optimized? How do the Authors know that this method provide good results?
Thank you for this comment. We added a sentence in section 4.3: Implemented lines 324-326
- Materials and method section – par. 4.3, 4.4 – determine the purity of used solvents (methanol, ethanol, chloroform, acetonitrile, formic acid, THF),
Thank you for your comment, we added a sentence: correction done line 322
lack of MS parameters (temperatures, potentials, collision energies, parent ions, method optimization etc.)
To the best of our knowledge, we have presented all the parameters involved in our MALDI-ToF MS analyses. The ion source is not heated and the potential of extraction is fixed for this type of instrument. Moreover, we only analyze our fingerprints in HRMS so we cannot provide you with the requested parameters such as collision energies, parent ions.
- Materials and method section – m/z 2 000 to 20 000 in the case of protein analysis and m/z 400 to 2 000 in the case of lipids analysis was used – why this m/z range was used? Did the Authors optimized the m/z range in the compounds identification? Is there any influence of the m/z range on the identified compounds, their nature and number?
Proteins are large biological molecules so we use a higher and larger mass range, it is the opposite for lipids which hardly exceed 1000 Da, at least in our experimental conditions. These informations are described in : Sandrin, T.R.; Goldstein, J.E.; Schumaker, S. MALDI TOF MS Profiling of Bacteria at the Strain Level: A Review. Mass Spectrom. Rev. 2013, 32, 188-217, doi:10.1002/mas.21359. It must be noted that only singly charged ions are formed by MALDI (compared to ESI where multiply charged ions are generated).
- how sensitive was the analysis?
It is not possible to give a precise answer to this question as the sensitivity depends on the molecules detected and we perform global analyses. Nevertheless the fingerprint was registered for all microorganisms using a single Petri dish.
- how many times was one sample analyzed?
Three times to confirm the fingerprints.
Check and correct the reference style according to the journal guide – positions 40, 41
Correction done after implementation
Make changes in the text.
Check and correct English
Done

Reviewer 2 Report
Specific comments concerning the manuscript entitled “Classification of Environmental Strains from Order to Genus Levels Using Lipid and Protein MALDI-ToF Fingerprintings and Chemotaxonomic Network Analysis” under reference Microorganisms-1595918.
I am somewhat divided over this manuscript. In one hand, this is a fact that mass spectrometry (MALDI-ToF MS here) associated to strong statistical analysis has proven to be one of the most powerful tools in the identification of microorganisms in many applications and the results displayed here for environmental strains are encouraging (as mentioned by the authors themselves). On the other hand, the results also may give cause for doubts concerning a successful classification of environmental strains. If the clustering by t-SNE of pathogenic bacteria (166 strains) from a part of the RKI data set looks very convincing, the following results show an increasing number of isolate nodes and apparition of illogical attributions within the clusters. This is particularly true for fungi.
In addition, the manuscript is often unclear because the many given values referring to figures seems to be wrong or are worthy of comments. For example, the caption of Figure 2 indicates 568 protein fingerprints but the sum of all fingerprints taken into account in the section 2.1., including the isolated nodes, is 509. In the same way, in paragraph 2.2., there are 44 isolated nodes coming from BNSB’s database but also 2 dark blue dots coming from RKI’s database. In figure 1, they were 5 and now in figure 2, only 2! Why? Furthermore, the number of isolated nodes is sometime wrong. For example in figure 4 (but not exhaustive), it is written that the set of isolated nodes contains 6 Actinomycetales (but there are 7 brown dots), 10 Bacillales (but there are 11 light green dots), 15 Burkholderiales (right), 10 Enterobacterales (but there are 11 dark blue dots), 1 Pseudomonadale, 1 Rhodospirillale and 1 Xanthomonadale (right for these last three genera). Another example of source of confusion concerns the Table S6 which indicates the parameters used for constructing the chemotaxonomic networks on MetGem for 5 figures whereas there are a total of 6 figures in the manuscript. but I think that the first line refers to Figure 2 (Figure 1 are mass spectra and not a t-SNE graph), line 2 to figure 3, etc. Please check these issues
Other points:
- Line 298: “phytopathogen” (“n” is missing)
- Line 344: replace “are” by “is”.
- Table 6S: I supposed that a discussion on the parameters used on MetGem is not the heart of this manuscript but few of these parameters intrigue me, regardless the figure number offset. The m/z tolerance on protein is higher than on lipids, which is easy to understand with ToF MS. However, why m/z is 4 for proteins in Figure 5 whereas it is only 2 for Figures 2 and 3? How the change in the cosine score is justified? Why is the perplexity value equal to 16 whereas it is equal to 11 for other figures?
To conclude, I am obviously not against the publication of this manuscript but I think it some issues deserve to be clarified before.
Author Response
Reviewer 2
Specific comments concerning the manuscript entitled “Classification of Environmental Strains from Order to Genus Levels Using Lipid and Protein MALDI-ToF Fingerprintings and Chemotaxonomic Network Analysis” under reference Microorganisms-1595918.
I am somewhat divided over this manuscript. In one hand, this is a fact that mass spectrometry (MALDI-ToF MS here) associated to strong statistical analysis has proven to be one of the most powerful tools in the identification of microorganisms in many applications and the results displayed here for environmental strains are encouraging (as mentioned by the authors themselves). On the other hand, the results also may give cause for doubts concerning a successful classification of environmental strains. If the clustering by t-SNE of pathogenic bacteria (166 strains) from a part of the RKI data set looks very convincing, the following results show an increasing number of isolate nodes and apparition of illogical attributions within the clusters. This is particularly true for fungi.
In addition, the manuscript is often unclear because the many given values referring to figures seems to be wrong or are worthy of comments.
- For example, the caption of Figure 2 indicates 568 protein fingerprints but the sum of all fingerprints taken into account in the section 2.1., including the isolated nodes, is 509.
Corrected line 124, rare orders not clustered together were not quantified before your comment. They represent 55 scattered protein fingerprints. However, the set of nodes described in section 2.1 constitutes a set of 513 fingerprints.
- In the same way, in paragraph 2.2., there are 44 isolated nodes coming from BNSB’s database but also 2 dark blue dots coming from RKI’s database. In figure 1, they were 5 and now in figure 2, only 2! Why?
There are only 2 isolated nodes left from the RKI spectral bank as these have clustered with spectral fingerprints from RKI and BSNB. The addition of data helps clustering of nodes due to spectral similarities. The t-distributed stochastic neighbor embedding (t-SNE) algorithm has been proven to outperform linear or nonlinear dimensionality reduction methods in several life-science research domains in terms of dimensionality reduction by catching local similarities in the high-dimensional space while trying to preserve the global structures as much as possible.
We have taken into account your comment by adding an explanatory sentence to the t-SNE algorithm: implemented lines 89-9. And adding this observation in section 2.2 : implemented lines 152-154
- Furthermore, the number of isolated nodes is sometime wrong. For example in figure 4 (but not exhaustive), it is written that the set of isolated nodes contains 6 Actinomycetales (but there are 7 brown dots), 10 Bacillales (but there are 11 light green dots), 15 Burkholderiales (right), 10 Enterobacterales (but there are 11 dark blue dots), 1 Pseudomonadale, 1 Rhodospirillale and 1 Xanthomonadale (right for these last three genera).
Fig. 2 and 3: Verified and implemented
Figure 4: Correction done lines 180-181
Figure 5: Implemented lines 209-212
Figure 6: Correction done line 234-235
- Another example of source of confusion concerns the Table S6 which indicates the parameters used for constructing the chemotaxonomic networks on MetGem for 5 figures whereas there are a total of 6 figures in the manuscript. but I think that the first line refers to Figure 2 (Figure 1 are mass spectra and not a t-SNE graph), line 2 to figure 3, etc. Please check these issues
Correction done in SI
Other points:
- Line 298: “phytopathogen” (“n” is missing)
Correction done line 304
- Line 344: replace “are” by “is”.
Correction done line 353
- Table 6S: I supposed that a discussion on the parameters used on MetGem is not the heart of this manuscript but few of these parameters intrigue me, regardless the figure number offset. The m/z tolerance on protein is higher than on lipids, which is easy to understand with ToF MS. However, why m/z is 4 for proteins in Figure 5 whereas it is only 2 for Figures 2 and 3? How the change in the cosine score is justified? Why is the perplexity value equal to 16 whereas it is equal to 11 for other figures?
Thank you for this comment. As far as we know, there are no universal parameters for creating this type of figure. Molecular networks are still a young tool (their first application dates from 2016, see Mingxun Wang, Jeremy J. Carver, Vanessa V. Phelan, Laura M. Sanchez, Neha Garg, Yao Peng, Don Duy Nguyen et al. "Sharing and community curation of mass spectrometry data with Global Natural Products Social Molecular Networking." Nature biotechnology 34, no. 8 (2016): 828. PMID: 27504778). In addition, we are the first to use this representation to classify organisms chemotaxonomically and it requires some hindsights on how to handle the data.
Next sentences are a citation from: Wattenberg, M.; Viégas, F.; Johnson, I. How to Use t-SNE Effectively. Distill; 2016. (https://distill.pub/2016/misread-tsne/)
« [t-SNE] goal is to take a set of points in a high-dimensional space and find a faithful representation of those points in a lower-dimensional space, typically the 2D plane. The algorithm is non-linear and adapts to the underlying data, performing different transformations on different regions. Those differences can be a major source of confusion. »
- However, why m/z is 4 for proteins in Figure 5 whereas it is only 2 for Figures 2 and 3?
As shown in Figures 5 and 6, our results demonstrate that our methodology is not the most suitable for classifying fungi. With a more stringent m/z tolerance (therefore equivalent to 2 or less) we obtain the following results:
On the other hand, we obtain this figure with m/z tolerance = 4:
As it can be seen here, a larger m/z tolerance allows us to obtain a better chemotaxonomic resolution in this case. This is due to the fact that the quality of the fungal protein fingerprints is lower than those obtained from bacterial extracts, i.e. less peaks (see: Normand, A.-C., Cassagne, C., Gautier, M., Becker, P., Ranque, S., Hendrickx, M., and Piarroux, R. (2017). Decision criteria for MALDI-TOF MS-based identification of filamentous fungi using commercial and in-house reference databases. BMC Microbiol 17, 25). This is one of the reasons why researchers use in-house databases. Moreover, fungi are more complex organisms than bacteria, i.e. multicellular. Thus, they are able to form sexual structures that could influence the profiles obtained modifying the intra-species and therefore inter-species taxonomic resolution due to this type of representation. To our knowledge, no study has addressed this issue, but here we explore the use of a global approach in contrast to most relative studies that focus on closely related taxa (see : Bader, O. (2017). Fungal Species Identification by MALDI-ToF Mass Spectrometry. In Human Fungal Pathogen Identification: Methods and Protocols, T. Lion, ed. (New York, NY: Springer), pp. 323–337.). As we said, these results demonstrate that our methodology needs to be improved to classify environmental fungi.
- How the change in the cosine score is justified?
In the same way as in the previous argument. Indeed, the calculation of cosine scores and thus the clustering of nodes will be largely influenced by the quality of the data and the quantity of nodes present. Here, the quality of the data is justified by Fig. 1, moreover we describe our method to obtain good quality MS1 spectra thanks to the R script (4.2.6) following the acquisition. Each fingerprint was manually checked upstream, i.e. peak intensity higher than the matrix and majority peaks with a S/N ratio > 6. In fact, we performed more than 780 protein and lipid fingerprints but only those that met the above criteria for both protein and lipid fingerprints were retained. If an extract from an isolate did not meet these criteria, the isolate was removed from the dataset.
The largely limiting factor of our study is the number of data acquired. In addition, some isolate fingerprints are over-represented, leading to illogical clustering with under-represented isolate fingerprints. Our approach is therefore applicable to sampling campaigns which also result, most of the time, in diversity bias (culture) but also in the representation of certain taxa (natural abundance of certain OTUs depending on the biotope studied).
Indeed, our methodology is "only" medium throughput. However, since the t-SNE is an iterative algorithm, it is possible to obtain better results by adding additional data. Indeed, the calculation of the cosine score is based on fingerprint similarities and not by discrimination. This is justified by the results obtained with the RKI database.
- Why is the perplexity value equal to 16 whereas it is equal to 11 for other figures?
« Perplexity is a measure for information that is defined as 2 to the power of the Shannon entropy. The perplexity of a fair die with k sides is equal to k. In t-SNE, the perplexity may be viewed as a knob that sets the number of effective nearest neighbors. It is comparable with the number of nearest neighbors k that is employed in many manifold learners. » Laurens van der Maaten (https://lvdmaaten.github.io/tsne/)
Perplexity is a modifiable parameter of t-SNE that balances the attention paid between local and global distance of the data. The performance of the t-SNE algorithm is rather robust depending on the requested perpexity. The most appropriate value depends on the density of the processed data. Most data sets will be processed with a perplexity between 5 and 50, according to Laurens van der Maaten. The more variables a dataset contains the more it would be necessary to increase this value. What matters most is to iterate until a stable configuration is obtained. This is the case here, with the maximum number of iterations allowed by the MetGem software, i.e. 10 000.
We understand your point. However, once a stable configuration is obtained it is possible to increase the perplexity without any influence on the representation, up to a certain threshold. The perplexity of this figure has only been modified for aesthetics.
As a proof of the non-influence of this parameter once a stable configuration is obtained, you will find below the original figure of the article (Fig. 5 - perplexity = 16 and the same figure with a perplexity = 11). If you think that the modification of this parameter for this one figure is troublesome, we can replace it with the generated one.
Figure 5 perplexity = 16
see attached document
Figure 5 perplexity = 11
To conclude, I am obviously not against the publication of this manuscript but I think it some issues deserve to be clarified before.

Round 2
Reviewer 1 Report
The revised version of the manuscript entitled "Classification of environmental strains from order to genus levels using lipid and protein MALDI-ToF fingerprintings and chemotaxonomic network analysis" (microorganisms-1595918), by Levasseur and co-workers meets all of my requirements. Authors gave valuable comments and answers for my questions. In this form, the presented manuscript can be accepted for publication.